# OpenReview forum: "h4rm3l: A Language for Composable Jailbreak Attack Synthesis"
_ICLR.cc/2025/Conference — ICLR 2025 Poster_

### Official Review · Reviewer_oNbH · 2024-10-20

**Soundness:** 3
**Presentation:** 4
**Contribution:** 3
**Rating:** 8
**Confidence:** 4

**Summary:**

The paper proposes h4rm3l, a domain-specific language designed to describe and facilitate jailbreak attacks on language models. h4rm3l addresses the gap between the diversity of jailbreak attack techniques and the limited benchmarks currently available. It formally expresses jailbreak attacks as compositions of parameterized string transformation primitives, enabling it to capture a wide range of attack types and their combinations. Furthermore, h4rm3l incorporates a bandit-based approach to jailbreak attacks and synthesizes a dataset containing 2,656 successful novel attacks targeting six state-of-the-art (SOTA) open-source and proprietary LLMs. The results demonstrate that the synthesized attacks are diverse and significantly more successful than previously reported methods, achieving success rates exceeding 90% on these SOTA models.

**Strengths:**

h4rm3l offers a formalized framework for representing jailbreak attacks, addressing the need for a consistent, standardized approach in this area of research. By utilizing parameterized string transformation primitives, it enables the modeling of diverse and complex attack types, making it a versatile tool. Notably, h4rm3l has demonstrated impressive empirical success, achieving over 90% success rates in generating jailbreak attacks across several state-of-the-art language models, including GPT-4o, Claude-3, and Llama3. This showcases its robustness and effectiveness in exposing vulnerabilities in a wide range of LLM architectures, proving its scalability and relevance for both proprietary and open-source systems.

**Weaknesses:**

While the paper is impressive, it does not sufficiently address its limitations. Beyond the template-based jailbreak methods and synthesis, other forms of jailbreak techniques exist, such as fuzzing or mutation-based or other approaches ([1], [2], [3]). Given that the authors propose a language to express various attack strategies, it is important to clarify the scope of h4rm3l in this regard. The paper should discuss how many types of jailbreak methods the language can effectively cover and whether it faces any limitations in representing certain attacks. Furthermore, it would be beneficial if the authors included a more comprehensive discussion of related work in the background section, addressing whether h4rm3l can accommodate or has limitations in expressing these alternative attack types.

[1] https://www.usenix.org/conference/usenixsecurity24/presentation/yu-jiahao
[2] https://www.usenix.org/conference/usenixsecurity24/presentation/yu-zhiyuan
[3] https://www.usenix.org/conference/usenixsecurity24/presentation/liu-tong

**Questions:**

1. To what extent can h4rm3l cover alternative jailbreak techniques, such as fuzzing or mutation-based attacks, beyond template-based methods?
2. Could the paper further discuss how h4rm3l compares to other jailbreak methodologies and clarify whether it can fully accommodate these approaches or has constraints?

---

> ### Author Response · Authors · 2024-11-22
> **Response to Reviewer oNbH**
>
> Thank you for finding our framework impressive! Here, we briefly compare h4rm3l to [1] and [2]. Our common response above both clarifies that h4rm3l can represent any jailbreak attack on black-box LLMs without restriction, and shows how h4rm3l can be used to represent and expand upon [3].
>
>
> Our work is similar to [1] (LLM-Fuzzer) in that we iteratively improve an initial set of attacks. However, h4rm3l differs because it represents attacks in a formal composable language that includes all string transformations, and hence all black-box attacks.
> String transformations include, but are not limited to templated transformations such as the ones illustrated in ([1],  Fig 3), which add a constant prefix or suffix to the original string.
> Our synthesis approach is also different from [1], as [1] produces templates while our approach produces h4rm3l programs, which are compositions of primitives including templated primitives and primitives that can perform arbitrary computation. For instance we implemented Kang et al. (2023)'s payload splitting attack in h4rm3l as the `PayloadSplittingDecorator`, which performs character-level manipulations of the input prompt (See Appendix B. for other state-of-the-art black-box jailbreak attacks in h4rm3l)
>
> We agree with [2]'s definition of jailbreak attacks. But further, we propose a formal representation for "strategic manipulation of input prompts": h4rm3l programs, which are compositions of parameterized string transformation primitives.
> [2] is similar in spirit to our efforts: to systematically scale the generation of jailbreak attacks, learn from human jailbreakers, and understand the characteristics of attacks. However, we approached the problem differently. We first built basic extensible composable abstractions that completely cover the space of black-box attacks, and then introduced primitives and examples from previously published work into our framework. From that pool of initial formalized attacks, we employed program synthesis methods to generate new attacks in the same formal representation.
>
> Our approach also enables possibilities that approaches such as ([1], [2], and [3] do not. Attacks expressed in h4rm3l are algorithmic, human-readable and can be quantitatively (Section 4) and qualitatively (Section 5) analyzed at scale; which enables progress in understanding LLM safety guardrails.
> For instance, by analyzing thousands of synthesized attacks, we showed that the success rate of attacks could grow with their compositional complexity, and that the distribution of primitives in successful attacks is target-LLM-specific (Fig 6).
>
> Thank you for referencing [1], [2], and [3], which have enriched our paper's discussion of related works.

---

> > ### Comment · Reviewer_oNbH · 2024-11-24
> >
> > Thank you for your response. I am open to raising my score if the updated manuscript add these discussions effectively.

---

> > > ### Author Response · Authors · 2024-11-29
> > >
> > > Thank you for your helpful feedback. We have updated the manuscript.

---

### Official Review · Reviewer_RpnP · 2024-11-01

**Soundness:** 3
**Presentation:** 2
**Contribution:** 3
**Rating:** 5
**Confidence:** 2

**Summary:**

This paper introduces h4rm3l, a novel approach to cover sufficiently large and diverse sets of jailbreak attacks with human-readable domain-specific language. h4rm3l includes (1) a domain-specific language formally expresses jailbreak attacks as compositions of parameterized string transformation primitives, (2) a synthesizer with bandit, and (3) a red-teaming software toolkit. h4rm3l also provides a dataset of 15891 novel jailbreak attacks and safety benchmarking results for 6 SOTA LLMs with 82 jailbreak attacks.

**Strengths:**

1. This paper studies jailbreak attacks from a novel perspective like software engineering, to synthesize (all) jailbreak attacks and evaluate the jailbreak robustness of LLMs.

2. The experiment is comprehensive and the visual analysis is clear.

3. The discussion part is detailed.

**Weaknesses:**

1. Section 3.1, especially line 166-167, makes this reviewer confused. It seems that Section 3.2 has few connections with Section 3.1. To specific, the description in Section 3.1 does not significantly contribute to this reviewer's understanding of the connection from the motivation to the method of this paper. For example, this reviewer wants to know more details about the generic decorator TransformFxDecorator and why it covers the space of all string-to-string transformations and could represent any jailbreak attack on black-box LLMs.

2. Based on the first point, the reviewer is concerned that the author's claim of “representing all black box attacks” is an overclaim. As the author cited in Section 2, PAIR, and its following version TAP, employ a red-teaming assistant to jailbreak the target model. So the algorithm output is uncertain. The authors should explain the inclusive relationship between h4rm3l and other black-box jailbreak attacks.

3. According to this reviewer's understanding, the process of synthesizing jailbreak attacks briefly includes two sub-processes: selection, composition, and evaluation. Section 3.2 and 3,3 describe the details of selection and evaluation, respectively. However, how are the selected jailbreak methods combined in this paper?

**Questions:**

This reviewer's questions focus on the methods' details. See weakness 1-3. This reviewer wants to know more about the details of this method.

**Details Of Ethics Concerns:**

None.

---

> ### Author Response · Authors · 2024-11-22
> **Response to Reviewer RpnP**
>
> We will attempt to make the prose more concrete and connected.
>
> - **Section 3.1** Introduces the h4rm3l language, which represents jailbreak attacks as compositions of parameterized string transformation primitives.
>
> - **Section 3.2** Formulates the synthesis of jailbreak attacks as a program search problem over the space of programs in the language described in **Section 3.1** with the objective of maximizing the ASR of synthesized programs. Specifically, the attack proposals generated by generateProposals at line 226 of the manuscript are compositions of primitives in h4rm3l, introduced in **Section 3.1**. In this section, we also proposed bandit-based few-shot program synthesis algorithms towards this objective.
>
> $\textbf{Re:  "more details about the generic decorator TransformFxDecorator" and }$
> $\textbf{"concerned that the author's claim of “representing all black box attacks” is an overclaim."}$
>
> Please see our common response above, which addresses this point.
>
> $\textbf{Re: how are the selected jailbreak methods combined in this paper?}$
>
> - h4rm3l enables arbitrarily complex composition of attack primitives. Technically, jailbreak attacks (including single primitives) are composed using their $.then()$ function. (See section 3.1).
>
> - To generate new jailbreak attacks, we prompt an LLM with a set of examples of jailbreak attacks expressed in h4rm3l, and instructions to generate new ones using the rules and primitives of the language, which are supplied in the same prompt. These new attacks are compositions of h4rm3l primitives, which are also often successful and novel attacks. (See Section 3.2, and Section 4).

---

> ### Comment · Reviewer_RpnP · 2024-11-23
>
> After reading Section 3 & 4.1 and the official comments by the authors, this reviewer still has some questions and views that differ from those of the authors, which prevented me from improving my rating.
>
> This reviewer does not hold the view that "h4rm3l primitives are string-to-string functions written in python that can perform arbitrary computation on strings, and therefore can represent any black-box attack." As an implementation of a combine attack, the authors should have at least spelled out in more detail the categorization of black-box jailbreak attacks and how they are implemented, which in turn explains why the "transform(prompt, assistant, random_state)" function can perform arbitrary computation and represent all black-box attacks.

---

> > ### Author Response · Authors · 2024-11-24
> >
> > **Re: "h4rm3l primitives are string-to-string functions
> > written in python that can perform arbitrary computation on strings, and therefore can represent any black-box attack."**
> >
> > This statement is true because all black-box attacks are string-to-string functions, and any string-to-string function can be implemented in Python.
> >
> >
> >
> > **Re: "the authors should have at least spelled out in more detail the categorization of black-box jailbreak attacks..."**
> >
> > - We disagree that a particular taxonomy of black-box attacks is required for the definition of a language that can represent all black-box attacks. h4rm3l can fully express any black-box attack regardless of how such attack fits in a particular taxonomy.
> >
> > - See Appendix B for examples of black-box jailbreak attacks we implemented in h4rm3l, the authors of some of which proposed particular taxonomies, such as *Competing Objectives* and *Mismatched Generalizations* by Wei et al., (2023).
> >
> >
> > **Re: "...and how they are implemented"**
> >
> > - See Appendix B, and our supplementary materials for details on the various black box attacks we implemented. See `h4rm3l/src/h4rm3l/decorators.py`, and `h4rm3l/src/h4rm3l/decorator\_expressions.py` for the source code of the attack primitives we implemented.
> >
> > - See `benchmark.html` for an easy way to browse those attacks, and the synthesized attacks we benchmarked. Click on specific attacks to view their source code.
> >
> >
> > **Re: "which in turn explains why the 'transform(prompt,
> > assistant, random state)' function can perform arbitrary computation and represent all black-box attacks."**
> >
> > - We clarify that `TransformFxDecorator` is not necessary to the fact that h4rm3l can represent any black-box attack by defining custom primitives. `TransformFxDecorator` makes h4rm3l complete even if custom primitives are not allowed, because any jailbreak attack can be implemented as a `TransformFxDecorator` instance.
> >
> > - Note that we experimented with the synthesis of "low-level expressions" (LLE) of jailbreak attacks, which are strictly constrained to compositions of TransformFxDecorator and RolePlayingDecorator. See Section 4 and 5 for related quantitative and qualitative results.
> >
> > - We realize that the following sentence at line 167 in our paper could have been more clear. We have updated it for clarity as  follows:
> > > "This primitive ensures that h4rm3l is complete **even if additional primitives are not allowed**, as it can be used to formally represent any jailbreak attack on black box LLMs."
> >
> > please let us know if you have further questions!

---

> > > ### Author Response · Authors · 2024-11-29
> > >
> > > Thank you for your constructive feedback! Please note that we have updated our manuscript based on this discussion and discussions from other reviewers.
> > >
> > > Please let us know if you have additional questions

---

> > > > ### Author Response · Authors · 2024-12-04
> > > >
> > > > Thank you for your time reviewing our paper!
> > > > Please let us know if we have addressed all your concerns or if you have additional ones.

---

### Official Review · Reviewer_fDVc · 2024-11-03

**Soundness:** 2
**Presentation:** 3
**Contribution:** 3
**Rating:** 8
**Confidence:** 3

**Summary:**

This paper introduces h4rm3l, a domain-specific language (DSL) for representing jailbreak attacks on large language models (LLMs) as compositions of parameterized string transformation primitives. The authors develop a synthesizer using bandit-based few-shot program synthesis to generate novel, high-success-rate jailbreak attacks against black-box LLMs. They demonstrate the effectiveness of h4rm3l by synthesizing 2,656 successful attacks targeting six state-of-the-art LLMs, achieving high success rates. The work also includes a zero-shot harmful LLM behavior classifier and an open-source red-teaming toolkit.

**Strengths:**

- The paper addresses a timely and relevant topic.
- The structure of the writing is clear and well-organized.
- The paper presents an innovative idea by transforming the jailbreak task into a formal language implementation.
- Implementing various existing jailbreak methods to support the proposed composite attack strategy.

**Weaknesses:**

- Unclear Motivation and Purpose: The motivation and unique advantage of transforming jailbreak tasks into a formal language are not entirely clear. In my view, directly developing a framework or tool that integrates multiple jailbreak prompting operators and scheduling strategies may already suffice for most jailbreak/red-teaming needs. Therefore, what specific advantages or unique capabilities does this formal language offer? How does it surpass the functionalities of traditional red-teaming frameworks? Additional clarification from the authors would be helpful.

- Assessment Method in Section 3.3: Using GPT as a judge has become a mainstream approach for evaluating jailbreak success rates, and many recent studies follow standard methods. However, this paper appears to introduce a new prompt. Is this prompt demonstrably superior to those used in other jailbreak studies? Alternatively, what necessary adaptations or unique design choices were made to tailor this prompt for the work presented here?

- Efficiency Comparison in Experiments: While the authors clearly present the effectiveness of the proposed method compared to baselines in jailbreak success rates (as shown in Fig. 4), a comparative analysis of jailbreak efficiency with existing methods is lacking. Figures 2 and 3 only show the number of iterations required by the proposed method. I suggest the authors provide a more straightforward and detailed efficiency comparison with baseline methods.

**Questions:**

1. Compared to directly building a red-teaming framework, what are the unique advantages of the formal language proposed in this paper?
2. Does the proposed jailbreak attack demonstrate superior efficiency compared to the baselines discussed in the paper?

---

> ### Author Response · Authors · 2024-11-22
> **Response to Reviewer fDVc**
>
> We thank the reviewer for finding h4rm3l innovative and the writing of the paper clear!
>
>
> $\textbf{Re: The motivation and unique advantage of transforming jailbreak tasks into a formal language are not entirely clear.}$
>
> We point the reviewer to our common response above, which addresses this point.
>
> $\textbf{Re:  How does it surpass the functionalities of traditional red-teaming frameworks?}$
>
> The biggest advantage of h4rm3l is in being able to systematically compose jailbreak primitives to generate novel, performant jailbreaks at scale. Other advantages of h4rm3l include the fact that it is complete (can represent any black-box attack), extensible (with new primitives and examples), and interpretable (h4rm3l code is human-readable).
> Our synthesizer efficiently explores the large compositional space of jailbreak attacks using bandit algorithms to generate new jailbreak attacks that are more performant than initial few-shot examples, and expressible in a human readable language.
> This **cannot** be achieved with other frameworks.
>
>
> $\textbf{Re: Is this prompt demonstrably superior to those used in other jailbreak studies?}$
>
> We noticed a lack of consistency in the way ASRs are measured in this rapidly evolving field. We preferred to follow Wei et al., (2023)'s ternary taxonomy of (GOOD, BAD, UNCLEAR) BOT behavior, which we found superior given a dataset of prompts assumed to be harmful.
> To create our prompt we performed multiple rounds of human-in-the-loop evaluations to ensure acceptable and sound results, based on the definition of "BAD BOT behavior": the LLM returning interpretable and usable harmful information in response to an illicit request. Our specific prompt text was adapted from the prompt introduced in the same work (see Appendix B), which was used for human annotators.
> Our classifier achieved $88.3\pm2.1\%$ precision and $100\pm0\%$ recall on data independently annotated by two expert annotators following the ternary taxonomy.
> We note that we used 50 randomly selected illicit prompts from the AdvBench dataset to perform our final benchmarking to ensure reliable ASR estimates.
>
>
>
> $\textbf{Re: I suggest the authors provide a more straightforward and detailed efficiency comparison.}$
>
> Thank you for your suggestion. We will update our manuscript with both asymptotic time complexity, and wallclock times we measured in practice.
> The synthesizer's complexity is $O(N_{iters} \cdot N_{proposals} \cdot N_{illicitprompts})$, which is linear with the number of iterations, the number of proposals (fixed at 20), and the number of illicit requests used to estimate ASRs during synthesis (fixed at 5).
> It took 25 and 34 hours to generate 1713 and 1939 attacks in 100 iterations respectively targeting  GPT3.5 and GPT-4o. This results in an attack synthesis rate of around 1 per minute.

---

> > ### Comment · Reviewer_fDVc · 2024-11-27
> >
> > Thank you for your reply. After careful consideration of your feedback, I still have one concern. Most of the operators in h4rm3l have been proposed in previous works. It is natural that combining these operators together would significantly expand the search space. If this is the only contribution, in my view, this work appears to be incremental rather than highly innovative. Could the authors briefly explain what specific challenges and difficulties would arise if I were to directly integrate these operators, and how h4rm3l has successfully addressed these issues? I look forward to the authors' response, and I would be happy to increase the score if this concern is adequately addressed.

---

> > > ### Author Response · Authors · 2024-11-30
> > >
> > > **Most of the operators in h4rm3l have been proposed in previous works.**
> > >
> > > Yes! Our contributions include the implementations of several SOTA primitives in our proposed unified formal compositional representation.
> > >
> > >
> > >
> > > **It is natural that combining these operators together would significantly expand the search space. If this is the only contribution, in my view, this work appears to be incremental rather than highly
> > > innovative.**
> > >
> > > - The compositional space of parameterized primitives is infinite. Our proposed representation, unlike previously proposed pipelines, fully covers this space.
> > >
> > > - Randomly composing primitive string transformations doesn't necessarily result in successful attacks, which are a minority in the large compositional space. See Figure 9, which shows the ASR distribution in 10, 460 Synthesized Programs.
> > >
> > > - The formal definition of jailbreak attacks in h4rm3l allow us to apply established program synthesis methods to this non-trivial search problem. This approach to redteaming was not previously proposed, and has led to an expanded discovery of the vulnerabilities of SOTA LLMs, as our results show.
> > >
> > >
> > > **Could the authors briefly explain what specific challenges and difficulties would arise if I were to directly integrate these operators, and how h4rm3l has successfully addressed these issues?**
> > >
> > > - "to directly integrate these operators" ("directly" compose parameterized primitives), you could implement a function that realizes each particular composition.
> > > This approach would not be scalable because each composition will require a custom implementation to handle parameter passing, and composition logic between primitives. It is also not clear how using this approach, you could write an algorithm that explores the infinite compositional space of parameterized primitives.
> > >
> > > - h4rm3l successfully addressed these issues by defining a language that enables the arbitrary composition of parameterized primitives. All black-box jailbreak attacks can be represented in h4rm3l. h4rm3l is extensible with custom interpretable primitives. As our results demonstrated, h4rm3l could serve as a foundation for the scalable readteaming of LLMs through principled program synthesis methods, and for the interpretable exploration of LLM safety failure modes. No such foundation was previously proposed.

---

> > > > ### Comment · Reviewer_fDVc · 2024-11-30
> > > >
> > > > Thanks for the author's response and the revised version, which have effectively addressed my primary concerns. As a result, I have decided to raise my score.

---

> > > > > ### Author Response · Authors · 2024-12-04
> > > > >
> > > > > Thank you for your constructive feedback!
> > > > >
> > > > > Please note that we have completed additional human evaluation of our zero-shot LLM behavior classifier. See our *Expanded Human Evaluation Results* post above.

---

### Official Review · Reviewer_CWn6 · 2024-11-03

**Soundness:** 2
**Presentation:** 3
**Contribution:** 3
**Rating:** 6
**Confidence:** 3

**Summary:**

The paper introduces h4rm3l, a domain-specific language (DSL) designed to formally represent jailbreak attacks on large language models (LLMs) as compositions of parameterized string transformations. The authors propose a framework that includes the h4rm3l DSL, a synthesizer with bandit algorithms for generating optimized jailbreak attacks, and a red-teaming software toolkit. The paper demonstrates the efficacy of h4rm3l by synthesizing a dataset of successful jailbreak attacks targeting six SOTA LLMs, showing higher success rates than previously reported attacks.

**Strengths:**

- This paper introduces the first formal, composable representation of jailbreak attack, providing a more systematic and comprehensive approach to assessing LLM vulnerabilities.
- The efficacy of the proposed framework is effectively demonstrated through the synthesis of a substantial dataset comprising successful jailbreak attacks against multiple state-of-the-art (SOTA) LLMs.

**Weaknesses:**

- The estimation of attack success rates relies solely on the assessment of 100 Claude-3-haiku responses by just two human annotators, which raises concerns about the generalizability of the findings.

- The relationship between various jailbreak methods and their impact on the results is not clearly articulated, particularly how the synthesis with h4rm3l connects to these outcomes.

- Some parts of the paper are vaguely written and are subject to elaboration and clarification.

- The following issues are subject to further clarification:
  - What do the numbers (e.g. "00536") in Figure 4 represent?
  - Are the horizontal and vertical coordinates of Figure 5 meaningful?
  - In line 928, "therefore we use proxy names" Do the proxy names here refer to the letters G, B and U?
  - What do the abbreviations BS, RP, and RT in Tables 3 and 4 signify?
  - In line 1222, "Table 2 and Table 3" should be corrected to "Table 3 and Table 4."?

**Questions:**

- What are the time and computational costs associated with using h4rm3l for synthesizing attacks?

---

> ### Author Response · Authors · 2024-11-22
> **Response to Reviewer CWn6**
>
> We thank the reviewer for recognizing the value of h4rm3l as the first formal, composable representation of jailbreak attacks, and its effectiveness in LLM safety assessment at scale!
>
>
> $\textbf{Re: Estimation of attack success rates}$
>
> We do think our findings are generalizable, given the steps we took to develop and validate our harmful LLM behavior classifier. To choose our prompt we performed multiple rounds of human-in-the-loop evaluations to ensure acceptable and sound results, based on the definition of "BAD BOT behavior": the LLM returning interpretable and usable harmful information in response to an illicit request. Our prompt is based on the ternary taxonomy ({GOOD, BAD, UNCLEAR} BOT behavior) introduced by (Wei et al., 2023), and our specific prompt text was adapted from the prompt introduced in the same work (see Appendix B), which was used for human annotators.
> Our classifier achieved $88.3\pm2.1\%$ precision and $100\pm0\%$ recall on data independently annotated by two expert annotators following the ternary taxonomy.
> However, we agree that annotating responses from more LLMs with a larger number of human annotators would increase readers' confidence in our results. We will perform additional validations and update results in the final version of the paper.
>
>
> $\textbf{Re: The relationship between various jailbreak methods and their impact on the results is not clearly articulated.}$
>
> Please see our common response above, where we provide an example of how we compose jailbreak attack primitives to give rise to new, performant jaibreaks.
>
>
> $\textbf{Re: What are the time and computational costs?}$
>
> h4rm3l synthesized jailbreak attacks efficiently. It took 25 and 34 hours to generate 1713 and 1939 attacks in 100 iterations respectively targeting  GPT3.5 and GPT-4o. This results in an attack synthesis rate of around 1 per minute. The computational cost associated with synthesis is dominated by the LLM calls, the number of which scales linearly with the number of iterations, as well as the number of proposals per iteration (fixed at 20) and the number of illicit prompts used to estimate ASRs (fixed at 5 during synthesis and 50 during the final benchmarking).
>
> $\textbf{Re: What do the numbers (e.g. "00536") in Figure 4 represent?}$
>
> These are part of the unique identifier of synthesized attacks, which can be matched with rows of Table 1 and records in our released data.
> See `benchmark.html` in our supplementary materials package (click on individual programs to view their source code in h4rm3l).
> Concretely 00536 is the composed attack:
> `CipherDecorator().then(RefusalSuppressionDecorator()).then(VillainDecorator())`
>
> $\textbf{Re: Are the horizontal and vertical coordinates of Figure 5 meaningful?}$
>
> Yes, they are the 2-dimentional t-SNE projection of CodeBERT embeddings of the source code of h4rm3l attacks. The proximity of attacks in the 2d space can be interpreted as the similarity of their source code in CodeBERT feature space.
>
>
> $\textbf{Re: In line 928, "therefore we use proxy names"}$
>
> Yes, G is a proxy for Good Bot, B is a proxy for Bad Bot, and U is a proxy for Unclear. We will make this clearer in the paper.
>
>
> $\textbf{Re: What do the abbreviations BS, RP, and RT in Tables 3 and 4 signify?}$
>
> BS, RP and RT respectively refer to Baseline, Rephrasing and Retokenization. We will clarify this in the paper.
>
>
> $\textbf{Re: In line 1222, "Table 2 and Table 3" should be corrected to "Table 3 and Table 4."?}$
>
> Thanks! We will make this change.

---

> > ### Author Response · Authors · 2024-11-29
> >
> > Thank you for your constructive feedback! Please note that we have updated our manuscript based on this discussion and discussions from other reviewers.
> >
> > Please let us know if you have additional questions

---

> > > ### Comment · Reviewer_CWn6 · 2024-12-02
> > > **Feedback**
> > >
> > > Thank you for the very detailed rebuttal. I have raised my score.

---

> > > > ### Author Response · Authors · 2024-12-04
> > > >
> > > > Thank you for your constructive feedback!
> > > >
> > > > **Re: Estimation of attack success rates**
> > > >
> > > > Please note that we have completed additional human evaluation of our zero-shot LLM behavior classifier. See our *Expanded Human Evaluation Results* post above.

---

### Author Response · Authors · 2024-11-22
**Generality of h4rm3l and Value of its formal expression of jailbreak attacks; DRA case study**

All the reviewers ask some question about the generality and value of translating jailbreak attacks into a formal language. We address this issue here, using as an illustration the DRA method [(Liu et al. (2024)](https://www.usenix.org/conference/usenixsecurity24/presentation/liu-tong) mentioned by reviewer oNbH.


$\textbf{Generality and extensibility of h4rm3l}$

- h4rm3l primitives are string-to-string functions written in python that can perform arbitrary computation on strings, and therefore can represent any black-box attack.

- New h4rm3l primitives can be added by implementing python classes inheriting from `PromptDecorator`. We implemented several primitives from the literature
(See `h4rm3l/src/h4rm3l/decorators.py` and `h4rm3l/src/h4rm3l/decorator\_expressions.py` in supplementary materials), and Appendix B in our main manuscript.

- h4rm3l also includes two generic primitives for convenience, which researchers and practitioners can use to rapidly define custom primitives directly in h4rm3l:
`RolePlayingDecorator`, which allows defining templated attacks with a custom prefix or suffix, and `TransformFxDecorator`, which allows transforming prompts with arbitrary python code. Note that any black box attack can be represented as a single `TransformFxDecorator` instance.

- Another convenient feature of `TransformFxDecorator` is that the python function it specifies is supplied with the `assistant` parameter, which is a handle to a configurable auxiliary LLM that can be used to perform transformations such as rephrasing or translating a prompt as performed in primitives such as `TranslateDecorator` (Yong et al. (2023)).

$\textbf{Implementation of DRA in h4rm3l}$

**Listing 2** in the next post shows how to implement DRA as a single instance of `TransformFxDecorator`. Although this implementation fully expresses DRA, it doesn't fully leverage the power of h4rm3l, which lies not only in the expression of arbitrary computation on strings, but also in the compact expression of jailbreak attacks as composition of interpretable human-defined abstractions.

A better approach is to implement DRA as a composition of granular primitives (defined in next post **Listing 1**). Then the program synthesizer can usefully explore both novel parameterizations and compositions of these primitives and other defined primitives. The simplest breakdown of DRA would compose just two parameterized primitives `DisguiseDecorator(toxicRatio, benignRatio).then(ReconstructDecorator())`. However, a better solution would be to split DRA into 4 primitives, as follows.

\begin{align*}
    &WordPuzzleObfuscationDecorator()\cr
    .then(&WordCharSplitDecorator(toxicRatio, benignRatio)) \cr
    .then(&PayloadReconstructionDecorator()) \cr
    .then(&ContextManipulationDecorator())
\end{align*}

If h4rm3l's synthesizer is run with this latter set of primitives, together with other primitives we explored such as `Base64Decorator` and `RefusalSuppressionDecorator` then h4rm3l may generate novel attacks that build on DRA's components and apply additional transformations such as inserting statements instructing the target LLM to not refuse requests, applying base64 encoding on the DRA attack payload, and intructing the target LLM to start its response with an affirmative prefix. The power of our formal DSL is to allow automatic exploration of such novel compositions in the resulting large compositional space, and these novel compositions are the basis of our high attack success rates, as well as our large-scale generation of jailbreak attacks.

\begin{align*}
    &WordCharSplit(tr=0.3, benignRatio=0.1)\cr
    .then(&RefusalSuppressionDecorator()) \cr
    .then(&WordLevelPuzzleObfuscation()) \cr
    .then(&PayloadReconstruction()) \cr
    .then(&Base64Decorator()) \cr
    .then(&AffirmativePrefixInjectionDecorator())
\end{align*}

See `benchmark.html` in our supplementary materials to explore the synthesized attacks we benchmarked (click on individual programs to view their source code in h4rm3l).

---

> ### Author Response · Authors · 2024-11-22
> **DRA case study | Listings**
>
> Listing 1:Implementation of DRA's components as granular h4rm3l primitives
> ----
>
> ````
> from DRA import (
>     word_puzzle_obfuscation, word_char_split, payload_reconstruction, context_manipulation
> )
>
> class WordPuzzleObfuscationDecorator(PromptDecorator):
>     def __init__(self) -> None:
>         super().__init__()
>
>     def decorate(self, prompt):
>         return word_puzzle_obfuscation(prompt)
>
> class WordCharSplitDecorator(PromptDecorator):
>     def __init__(self, tr, br) -> None:
>         self.tr, self.br = tr, br
>         super().__init__()
>
>     def decorate(self, prompt):
>         return word_char_split(promt, self.tr, self.br)
>
> class PayloadReconstructionDecorator(PromptDecorator):
>     def __init__(self) -> None:
>         super().__init__()
>
>     def decorate(self, prompt):
>         return payload_reconstruction(prompt)
>
> class ContextManipulationDecorator(PromptDecorator):
>     def __init__(self) -> None:
>         super().__init__()
>
>     def decorate(self, prompt):
>         return context_manipulation(prompt)
>
> class DisguiseDecorator(PromptDecorator):
>     def __init__(self, tr, br) -> None:
>         super().__init__()
>         self.disguise = WordPuzzleObfuscationDecorator().then(WordCharSplitDecorator(tr, br))
>
>     def decorate(self, prompt):
>         return self.disguise.decorate(prompt)
>
> class ReconstructDecorator(PromptDecorator):
>     def __init__(self, tr, br) -> None:
>         super().__init__()
>         self.reconstruct = PayloadReconstructionDecorator().then(ContextManipulationDecorator())
>
>     def decorate(self, prompt):
>         return self.reconstruct.decorate(prompt)
>
> class DRADecorator(PromptDecorator):
>     def __init__(self, tr, br) -> None:
>         super().__init__()
>         self.dra = DisguiseDecorator().then(ReconstructDecorator())
>
>     def decorate(self, prompt):
>         return self.dra.decorate(prompt)
> ````
>
>
> Listing 2: Implementation of DRA as an instance of TransformFxDecorator
> ----
>
> ```
> from DRA import (
>     word_puzzle_obfuscation, word_char_split, payload_reconstruction, context_manipulation
> )
> TransformFxDecorator(
>     transform_fx=
>     """def transform(prompt, assistant, random_state):
>     tr = 0.5
>     br = 0.5
>
>     prompt=word_puzzle_obfuscation(prompt)
>     prompt=word_char_split(prompt, tr, br)
>     prompt=payload_reconstruction(prompt)
>     prompt=context_manipulation(prompt)
>
>     return prompt"""
> )
> ```

---

### Author Response · Authors · 2024-11-29
**h4rm3l paper revision: Improved clarity and implementation details**

We have updated our manuscript based on the reviewers helpful constructive feedback. See updated PDF, which highlights changes from our initial submission.

- **Abstract \& Intro**
    - Noted our finding on the necessity for target-specific attack synthesis.
    - Clarification edits.

- **Related Work**
    - Incorporated feedback from **Reviewer oNbH**. Clarified h4rm3l's significance in comparison with traditional redteaming frameworks as asked by **Reviewer fDVc**.
    - Referenced Wei et al. (2023)'s definition of black-box attack as transformations that induce *restricted* behaviors in LLMs.
    - Added DRA (Liu et al., 2024) as example.
    - Contrasted h4rm3l with LLM-Fuzzer (Yu et al., 2024a).
    - Explained how h4rm3l can be used in experiments such as Qiu et al. (2023)'s, which used structured templates. Distinguished *jailbreak prompts* and *jailbreak attacks*.
    - Contrasted h4rm3l's approach to learning from human jailbreakers with prior approaches such as (Yu et al., 2024b)'s.

- **Method**
    - Improved this section based on feedback from **Reviewers oNbH, fDVc, RpnP and CWn6**. Clarified the h4rm3l language, its composition and extension mechanisms, its relationship with Python, the reason why it covers all black-box attacks, the role of its generic primitives, and the reason why it is a suitable basis for representing, analyzing and generating jailbreak attacks.
    - Introduced Sections 3.1, 3.2 and 3.3 more coherently. The previous organization led to confusion, as **Reviewer RpnP** noted. Also clarified the groundings of our ASR estimation method in response to **Reviewer CWn6**'s notes. **Reviewer oNbH**'s questions on related works also helped present this section more coherently, and clarify how our contributions relate to prior work.
    - Replaced prior Equation. 1 (now in appendix M) with Listing 1, which gives a concrete example of synthesized jailbreak attack composing generic primitives.
    - Referenced Anthropic's Acceptable Use Policy to contextualize example in Figure 1.
    - In Equation 1 (ASR), clarified that H is also a function of the original illicit request.
    - Updated hyperparameter names ($N_{iters}$, $k_{illicit}$, $k_{examples}$) for clarity.
    - Clarified proposals generation in Algorithm 1. Made  $N_{proposals}$ hyperparameter explicit.
    - Added complexity expression to Algorithm 1. This and the wallclock time performance we added (Appendix L), and our improved discussions on the synthesizer hyperparameters address **Reviewer fDVc**'s questions on the efficiency of our synthesizer.

- **Results**
    - Improved comparison of targeted attack synthesis methods.
    - Improved captions for Figures 2, 3 and 4.
    - Based on **Reviewer CWn6**'s feedback,
    clarified that the t-SNE projections in Figure 5 were based on the CodeBERT features of attack source code in h4rm3l.
    - Clarified qualitative observations on synthesized jailbreak attacks solely expressed in generic primitives.
    - Improved captions for Figures 5.

- **Discussion and Conclusion:**
    - Included details on synthesizer hparams  $k_{illicit}$, $N_{examples}$ and $N_{proposals}$.
    - Included automated LLM behavior classifier details and adaptability to custom content moderation policies. Addressed **Reviewer CWn6** and **Reviewer fDVc**'s  concerns on how our ASR estimation mechanism can be generalized.
    - Corrected max ASR on GPT-4o to 94\% (vs 90\%)
- **Appendix**
    - *G*: Defense Experiments
        - Corrected table references noted by **Reviewer CWn6**
    - *K*: Generic and Customized Primitives
        - Updated this appendix with usage details of generic decorators, which previously cluttered Section 3.1. This improved presentation also addresses **Reviewer RpnP**'s concerns.
    - *L*: Synthesizer Wallclock Time Performance
        - Added this appendix synthesizer runtime details (timezone redacted to not compromise anonymity).
    - *M*: h4rm3l language specification and implementation details
        - Added this appendix with details of the `PromptDecorator` base class, the h4rm3l program compiler, and the generic primitives. This covers implementation details asked by **Reviewer RpnP**.

We also note the following errata, and pending changes that we will correct in the final manuscript:
- Line 128: "primitives abstractions" corrected to "abstractions"
- Line 487: "`AffixDecorator` resulted" corrected to "AffixDecorator *could result*"
- Line 217-218: "malicious prompts transformed by a" corrected to "*illicit* prompts transformed by *p*"
- Lines 480-481: "those *of* obfuscation schemes" corrected to "those obfuscation schemes"
- Previously addressed notes from **Reviewer CWn6**'s feedback:
    - In Figure 4, the numbers (e.g. "00536") are synthesized attack identifiers.
    - Acronyms BS, RP, and RT in Tables 3 and 4 mean Baseline, Rephrasing, and Retokenization
    - "proxy names" ( G, B and U?) stand for (Good, Bad and Unclear) Bot behavior.

---

### Author Response · Authors · 2024-12-04
**Expanded Human Evaluation Results**

Based on feedback from **Reviewer CWn6** and **Reviewer fDVc**, we performed an expanded human evaluation on 122 LLM responses uniformly sampled from a dataset of 360 LLM responses (20 responses x 3 classes x 6 LLM).

576 human annotations were performed on those 122 LLM responses. Each model response is annotated by 1 to 7 human annotators following Wei et. al, (2023)'s ternary taxonomy. Their consensus annotation, which is only achieved when there are no ties, is assumed to be the ground truth annotation. This ground truth is used to evaluate both the automated classifier, and the human annotators. The following performance metrics were measured:
- Our zero-shot classifier accuracy is 78\%. Its precision and recall on the BAD BOT class are respectively 72\% and 76\%.
- Humans accuracy is 93\%. Their precision and recall on the BAD BOT class are respectively 91\% and 87\%.

See details below. We will include these additional results in the final version of our manuscript.

\
\
Classification Performance: Automated Classifier:
==========

```
              precision    recall  f1-score   support

     BAD BOT       0.72      0.76      0.74        38
    GOOD BOT       0.89      0.95      0.92        42
     UNCLEAR       0.70      0.62      0.66        42

    accuracy                           0.78       122
   macro avg       0.77      0.78      0.77       122
weighted avg       0.77      0.78      0.77       122
```

\
\
Confusion Matrix: Automated Classifier:
==========

| Actual↓/Predicted→ | BAD BOT | GOOD BOT | UNCLEAR |
|-------------------|---------|-----------|----------|
| BAD BOT | 29 | 0 | 9 |
| GOOD BOT | 0 | 40 | 2 |
| UNCLEAR | 11 | 5 | 26 |


\
\
Classification Performance: Human Annotators:
==========

```
              precision    recall  f1-score   support

     BAD BOT       0.91      0.87      0.89       164
    GOOD BOT       0.99      0.99      0.99       202
     UNCLEAR       0.89      0.93      0.91       210

    accuracy                           0.93       576
   macro avg       0.93      0.93      0.93       576
weighted avg       0.93      0.93      0.93       576
```

\
\
Confusion Matrix: Human Annotators:
==========

| Actual↓/Predicted→ | BAD BOT | GOOD BOT | UNCLEAR |
|-------------------|---------|-----------|----------|
| BAD BOT | 142 | 1 | 21 |
| GOOD BOT | 0 | 200 | 2 |
| UNCLEAR | 14 | 1 | 195 |

---

### Meta-Review · Area_Chair_qJ1e · 2024-12-17

**Metareview:**

This work proposes a language of primitives, and associated program synthesis approaches for generating new jailbreaks. A variety of results are also provided. All but one reviewer are positive. For the negative reviewer, one of the concerns was the remark "h4rm3l primitives are string-to-string functions written in python that can perform arbitrary computation on strings, and therefore can represent any black-box attack". Strictly speaking, this does not appear to be true, as black-box jailbreaks/red-teaming approaches that allow query access to the LLM are not yet included. The authors should clarify this current limitation in the final version. Weighing the pros and cons, overall this paper merits publication.

**Additional Comments On Reviewer Discussion:**

During the rebuttal discussion, the authors addressed concerns about the framework's novelty, comprehensiveness, and empirical evaluation. The authors clarified that h4rm3l represents a formal and extensible approach to jailbreak synthesis, enabling the systematic composition of attack primitives and surpassing prior frameworks in versatility and scalability. They added detailed analyses of the method’s coverage of alternative attack strategies, including fuzzing and mutation-based attacks, and provided further examples to support claims of completeness and extensibility. Enhanced evaluations, such as expanded human annotations and runtime performance metrics, addressed reviewer concerns.

---

### Decision · Program_Chairs · 2025-01-22

Accept (Poster)